

# Deep learning in finance assessing twitter sentiment impact and prediction on stocks

Kaifeng Guo and Haoling Xie

Maynooth International Engineering College, Fuzhou University, Fuzhou, Fujian, China

## ABSTRACT

The widespread adoption of social media platforms has led to an influx of data that reflects public sentiment, presenting a novel opportunity for market analysis. This research aims to quantify the correlation between the fleeting sentiments expressed on social media and the measurable fluctuations in the stock market. By adapting a pre-existing sentiment analysis algorithm, we refined a model specifically for evaluating the sentiment of tweets associated with financial markets. The model was trained and validated against a comprehensive dataset of stock-related discussions on Twitter, allowing for the identification of subtle emotional cues that may predict changes in stock prices. Our quantitative approach and methodical testing have revealed a statistically significant relationship between sentiment expressed on Twitter and subsequent stock market activity. These findings suggest that machine learning algorithms can be instrumental in enhancing the analytical capabilities of financial experts. This article details the technical methodologies used, the obstacles overcome, and the potential benefits of integrating machine learning-based sentiment analysis into the realm of economic forecasting.

## INTRODUCTION

In the contemporary financial realm, artificial intelligence (AI) is reshaping trading and investing, playing a crucial role in handling vast financial data through pattern recognition, natural language processing, and predictive analytics. This tech evolution is notably visible as financial markets merge with the swift information flow, acknowledging platforms like Twitter as influential market drivers. Utilizing advanced AI algorithms for Twitter sentiment analysis has shown promise in deciphering market trends (*Xiao & Ihnaini, 2023*). In brand management and marketing, companies can leverage Twitter sentiment analysis to understand consumer perceptions of their brand and products. By monitoring sentiment on Twitter, they can swiftly identify and respond to negative sentiment while reinforcing positive sentiment, thereby improving brand reputation and increasing market share. Furthermore, monitoring public opinion on social media platforms through sentiment analysis to grasp its impact on stock prices signifies a notable advancement (*Ko & Chang, 2021*). While public sentiment influencing market trends is not new, AI's real-time data processing capabilities have ushered in a transformative era in financial analysis,

Corresponding author
Kaifeng Guo, 1362106037@qq.com

marking a significant juncture where AI and finance intersect, opening doors to innovative methodologies in stock market analysis.

The research to date still leaves something to be desired. Firstly, the predictive accuracy of social media sentiment in stock market movements is not consistent across all sectors and companies. As noted in *Geven*'s *(2019)* research, while individual tweets by influential figures like Donald Trump can impact specific company stocks, they do not necessarily affect broader market indices like the S&P 500. This inconsistency suggests that the predictive power of social media may be more nuanced and context-dependent than initially thought. Secondly, there is the challenge of noise and relevance in social media data. *Corea (2016)* studies highlight that the average sentiment of tweets may not be as predictive as the volume of tweets, underscoring the difficulty in filtering out noise and identifying relevant data. This is further complicated by the varied nature of social media discourse, which can be influenced by factors unrelated to market dynamics. Another limitation is the reliance on complex models and algorithms, which, while effective, may not be entirely transparent or understandable to all users. The studies by *Domeniconi et al. (2017)* and *Moro et al. (2019),* for example, demonstrate high prediction accuracy but rely on sophisticated text mining and machine learning techniques that may not be easily replicable or interpretable by less technical stakeholders. Additionally, the focus of existing research has primarily been on short-term predictions. The long-term impact of social media sentiment on stock markets remains less explored, raising questions about the sustainability and long-term reliability of using social media data for financial forecasting. Lastly, most of these studies are limited by their retrospective nature. They analyze historical data and trends, which may not necessarily predict future market behaviors, especially in the face of unprecedented events or shifts in market sentiment.

Our article addresses the limitations in existing research on the relationship between social media sentiment and stock market movements and proposes several strategies to overcome these challenges and advance the field. Firstly, to improve the predictive accuracy across different sectors and companies, we have fine-tuned the sentiment language model specifically for Twitter stock comments. This approach enables a better understanding of the sentiment in diverse market contexts, resulting in an improved correlation between sentiment analysis results and actual stock prices. Our model focuses on the specific domain of Twitter stock comments to capture the subtleties and variations in sentiment that are unique to this medium and its impact on stock markets. Secondly, recognizing the need to filter noise and enhance data relevance, our research includes extensive experiments across three datasets. These experiments rigorously test and validate the link between tweet sentiments and stock prices. Our comprehensive experimental approach aims to strengthen the evidence base for the predictive power of social media sentiment, contributing to more accurate and reliable stock price predictions. Furthermore, we have used a wide range of evaluation functions to analyze the outcomes of our experiments in response to concerns about the complexity and opacity of predictive models. This approach enhances the transparency and interpretability of our findings, ensuring that our results are accessible and understandable to a broader audience, including those with less technical expertise. By

doing so, we hope to bridge the gap between complex machine-learning techniques and practical applications in financial forecasting.

In summary, this article makes several contributions:

- We not only fine-tune the sentiment language model so that it can be applied to Twitter stock comments but also analyze the correlation between the results obtained and the stock price with the predicted.
- We conducted extensive experiments on the three datasets and demonstrated that the sentiment of tweets is indeed linked to stock prices, contributing to the accuracy of stock price predictions.
- We used a large number of evaluation functions to analyze the results obtained from a large number of experiments.

## RELATED WORK

The integration of AI and machine learning algorithms in financial markets has transitioned from an experimental approach to a more established strategy among traders and investors. Pioneering studies such as those by *Bollen, Mao & Zeng (2010)* have demonstrated the potential of social media sentiment in predicting stock market movements, showing that Twitter mood could predict the Dow Jones Industrial Average with an 87.6% accuracy. Following this, *Zhang, Fuehres & Gloor (2011)* further explored this domain, revealing the capability of Twitter sentiment to forecast the daily direction of stock prices with a significant level of confidence. These initial findings have set a foundation for deeper exploration into sentiment analysis and its practical implications in stock trading.

*Ranco et al.*'s *(2015)* investigation into the relationship between financial news and Twitter posts has reinforced the notion of a significant predictive relationship between the two, particularly during significant market events. This underscores the growing importance of parsing through vast amounts of unstructured data to uncover patterns and signals that may elude human detection.

Building on these concepts, *Mao, Wei & Wang (2013)* dissertation illustrates the effective use of Twitter data in analyzing stock market behaviors and aiding trading decisions, particularly demonstrating a significant correlation between tweet volumes and stock trading volumes for S&P 500 stocks. In a similar vein, the 2016 study by *Tan et al. (2016)* employed non-Gaussian SVAR to correlate Twitter sentiment with stock market movements, offering a nuanced approach compared to traditional models.

However, studies such as those by *Kiro (2014)* and *Geven (2019)* present more nuanced or specific scenarios. *Kiro*'s *(2014)* research, while successful in classifying tweet sentiments, did not find a significant correlation with financial trend indicators. *Geven*'s *(2019)* study, on the other hand, indicated that while Donald Trump's tweets did not generally affect the S&P 500 as a whole, they did have an observable impact on individual company stocks.

Further supporting the predictive power of social media sentiment, the studies by *Domeniconi et al. (2017)* and *Moro et al. (2019)* demonstrated high accuracy in predicting stock market movements. *Domeniconi et al. (2017)* achieved an 88.9% accuracy in predicting Dow Jones movements using a method based on text similarity measures and mining Twitter data, while *Moro et al. (2019)* developed a text mining method that diverges from traditional sentiment analysis, focusing on identifying relevant tweets for predicting DJIA movements.

The potential of AI in predicting stock prices based on social media sentiment has led to discussions on market efficiency and the implications of using algorithmically driven trading strategies. While *Sprenger et al. (2014)* suggests that AI's ability to anticipate market movements based on public sentiment could reduce market inefficiencies, *Lachanski & Pav (2017)* caution about the risks of creating echo chambers and potentially amplifying market volatility. This ongoing research highlights the dynamic interplay between digital sentiment and financial market fluctuations, illustrating both the opportunities and challenges in integrating AI into the financial sector.

# PROPOSED METHOD

This article introduces a novel approach to integrate social media sentiment with stock market data to predict stock price movements. Our methodology is comprised of two key components: a fine-tuned sentiment analysis model based on the RoBERTa (*Camacho-Collados et al., 2022*) architecture and a predictive recurrent neural network (RNN)-based model. The workflow is illustrated in Fig. 1 and consists of the following steps.

## Sentiment analysis

To begin, we gather a large number of tweets related to different companies that will serve as the fundamental basis for our sentiment analysis. We employ a pre-trained RoBERTa base model, which is known for its exceptional performance in natural language understanding tasks. This model is then further fine-tuned on a labeled Stock-Market Sentiment Dataset to tailor its predictions to the financial domain. During the fine-tuning process, the model's weights are adjusted to better interpret the language and sentiments expressed in stock market-related conversations.

Fine-tuning is carried out using a supervised learning approach where the model is exposed to examples of tweets paired with sentiment labels. The loss function is then optimized to reduce the discrepancy between the predicted sentiment and the actual sentiment label. The result of this phase is a sentiment analysis model that is capable of identifying the underlying sentiment in tweets regarding companies, classifying them as positive, neutral, or negative.

## Stock price prediction model

The approach we use involves two components. Firstly, we use a sentiment analysis model to determine the public sentiment towards a particular stock. Secondly, we use an RNN-based model to forecast stock price trends. RNNs are particularly suited for this task as

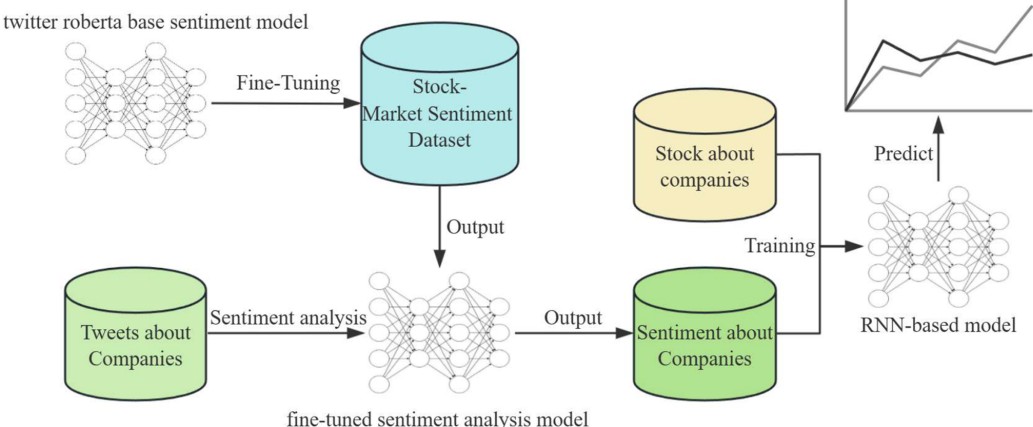

**Figure 1 An overview of the main approaches in our articles.**

they can maintain a memory of previous inputs, making them ideal for sequential data such as time series.

We train the RNN with a dataset comprising historical stock prices and the output sentiment from the sentiment analysis model. Through the training process, the RNN learns to recognize patterns and correlations between market sentiment and stock price movements. The objective of training is to minimize the error between the model's predictions and the actual stock price movements. The RNN's parameters are adjusted through backpropagation and optimization algorithms to improve its forecasting accuracy.

The integration of sentiment analysis output into the stock price prediction model is crucial. By incorporating sentiment data, the RNN model gains access to a broader context beyond mere price history, encompassing the public sentiment that can indicate future market behavior.

To predict future stock prices, the trained RNN model processes the latest sentiment analysis outputs along with recent stock price data. The model outputs predicted trends in stock prices, which are represented as a time series forecast.

## EXPERIMENTS

### Experimental setting

#### Datasets

In our study, we utilized three datasets, the first one depicted in Fig. 2, which was combined with binary sentiment scores. The dataset included a total of 5,791 text entries, each linked with a sentiment score indicating either positive (1) or negative (0) sentiment. The content of the 'text' variable encompassed market comments, opinions, and forecasts that reflected various themes related to the stock market. By using this data, we could utilize a pre-trained sentiment assessment model.

The second dataset that we have is focused on Twitter activity related to Tesla. This dataset consists of a total of 80,793 tweets that are associated with 25 different stocks. The purpose of this dataset is to investigate the correlation between public sentiment,

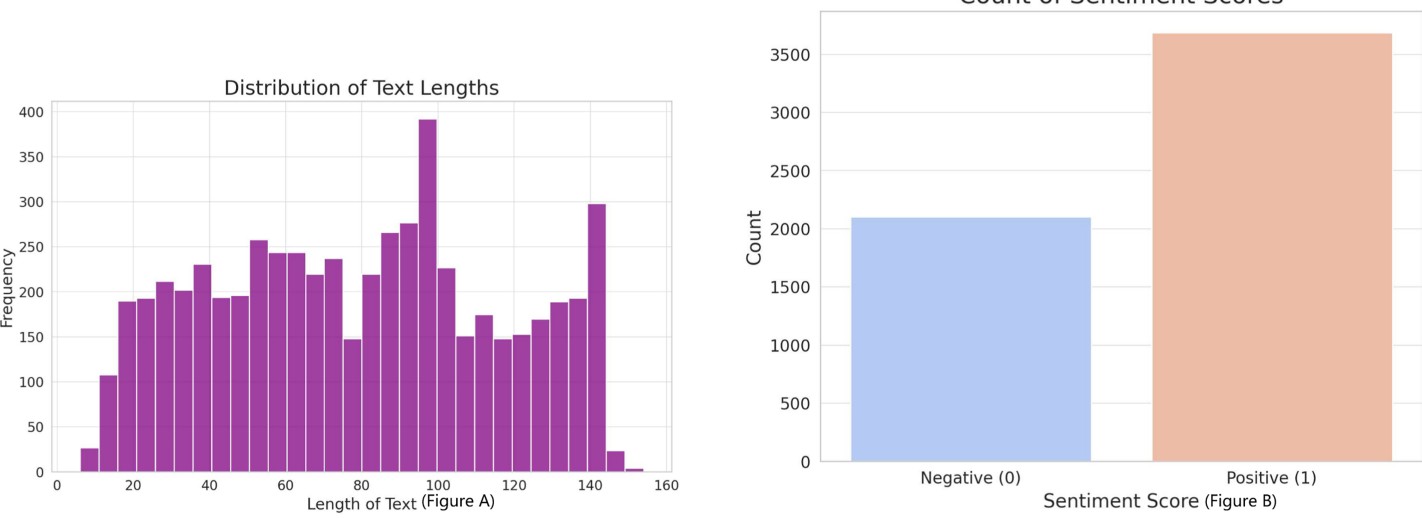

**Figure 2** (A) Distribution of sentiment scores and text lengths in the corpus. (B) Indicates a predominance of positive sentiment, while the histogram reveals a concentration of text entries within a specific length range, reflecting the dataset's compositional characteristics.

discussion volume, and market behavior. Figure 2 shows the distribution of tweet lengths related to this company. This distribution captures the variation and concentration of character usage in public discourse over time. The tweets were collected to capture a wide range of sentiments, from brief mentions to more elaborate discussions about the company. Figure 3 shows a closer look at the length distribution can provide insights into the level of engagement and information depth that Twitter users contribute to Tesla. This dataset not only includes tweet length but also time-series data. The left panel of Fig. 4 presents the trend of tweet volumes alongside the stock price movements post-normalization. This provides a temporal viewpoint of social media's influence on stock prices.

Table 1 displays the third dataset, which consists of tweet volumes for the top technology companies from 2015 to 2020. The dataset covers Apple, Amazon, Google Inc., Microsoft, and Tesla Inc., all of which are major players in their respective markets with significant public visibility. The tweet volumes act as a quantitative measure to assess public interest and sentiment towards these companies, forming a basis for comparative sentiment analysis across different entities within the tech sector.

Each dataset has a specific purpose in our analysis. The distribution of tweet length helps us understand how Twitter users talk about Tesla, and we can correlate the length of the tweets with the intensity of sentiment or informational content. The time series and volume data give us a dynamic view of how social media activity correlates with stock prices over time. Finally, the cross-company Twitter volume dataset allows us to compare sentiment analysis and suggests that higher tweet volumes may indicate more significant market movements or changes in sentiment.

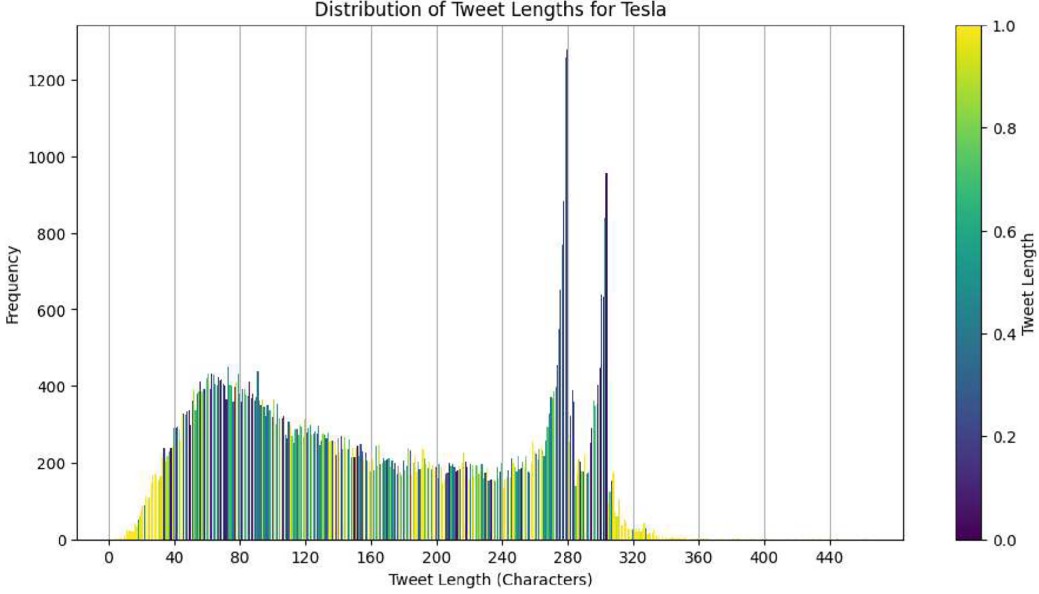

**Figure 3  Distribution of tweet lengths for tesla.**

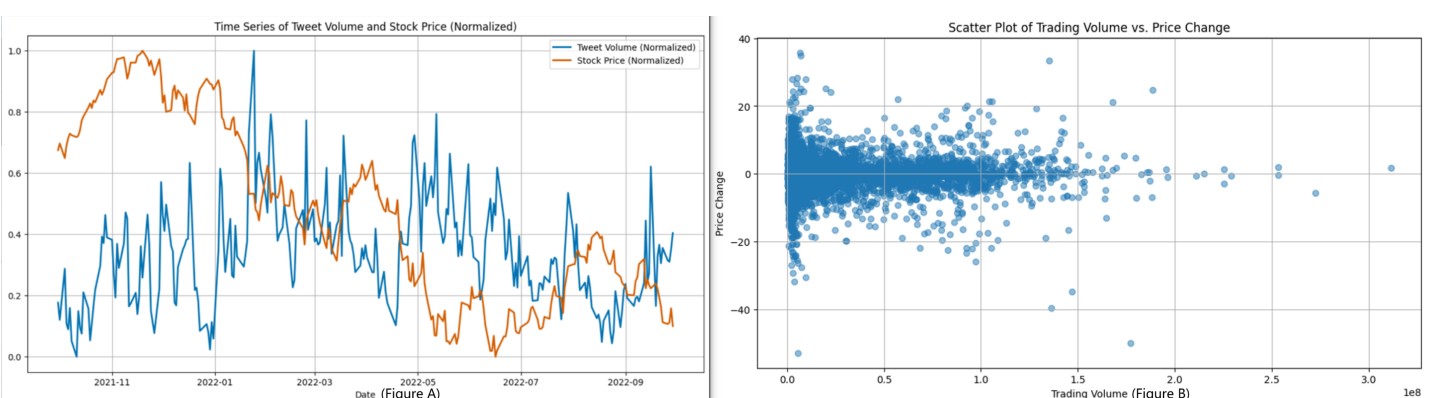

**Figure 4** (A) Shows the time-series trend of tweet volume and stock price after normalization in the dataset, and (B) is a scatter plot between trading volume and price change.

| Table 1  Tweets volume about the top companies from 2015 to 2020. | |
|---|---|
| **Company** | **Tweet** |
| Apple | 1,425,013 |
| Amazon | 718,715 |
| Google Inc. | 720,138 |
| Microsoft | 375,711 |
| Tesla Inc. | 1,096,868 |

Integrating these datasets enables us to explore the influence of social media sentiment on stock market performance comprehensively. We can examine not only the breadth of discussion but also its depth and temporal alignment with stock price fluctuations.

*Implementation*

In our experiments, we started by dividing the fine-tuning dataset into a training set and a validation set, where the validation set accounted for 10% of the entire dataset. To ensure consistency and reproducibility of the data segmentation, we set a random seed of 42. We performed the fine-tuning task based on the pre-trained model twitter-roberta-base-sentiment-latest (*Camacho-Collados et al., 2022*). This model has a hidden layer size of 768 and an intermediate layer size of 3,072, with 12 attention heads and 12 hidden layers. To maintain uniformity, all text sequences were truncated or patched to a length of 128 characters.

During the training process, we followed a strict parameter configuration. The number of training cycles was set to 10 epochs, with a training batch size of 16 on each device and an evaluation batch size of 64. We also set the warm-up step to 500 steps and the weight decay to 0.01. For the stock price prediction training period, we set the epoch to 150, the learning rate to 0.0001, and the window size to 2.

In the prediction task, we used an LSTM with a linear layer. The hidden layer of the LSTM was set to 256, and the linear layer outputted the predicted values. We used TSLA from September 2021 to June 2022 as the training set, and AAPL's April 2018 to August 2018 as the validation set, while the rest of the data was used as the test set.

## Evaluation metrics

Sentiment analysis of stock-related tweets commonly uses precision, recall, and the F1 score as metrics. These quantitative measures help to compare the predictive capabilities of various models or the same model at different stages of training.

Precision is a measure of how accurate the model's positive predictions are. It is defined as:

$$P = \frac{TP}{TP + FP}$$

where *TP* represents the number of true positives, and *FP* represents the number of false positives.

Recall measures the model's ability to identify all relevant instances correctly. The recall *R* is defined as:

$$R = \frac{TP}{TP + FN}$$

where *FN* represents the number of false negatives.

The F1 score is the harmonic mean of precision and recall, providing a balance between the two metrics. It is particularly useful when the class distribution is uneven. The F1 score reaches its best value at 1 (perfect precision and recall) and worst at 0. The *F1* score is defined as:

$$F1 = 2 \cdot \frac{P \cdot R}{P + R}$$

In the analysis of the fine-tuned sentiment analysis model, these metrics allow for a nuanced understanding of how well the model performs, particularly in the domain-specific context of stock market-related social media sentiment.

To calculate the value of the relationship between sentiment and stock price, we usually use the directional consistency percentage and the Pearson correlation coefficient.

$$\Delta \text{ sentiment}(t) = \text{sentiment}(t) - \text{sentiment}(t-1)$$

where sentiment(t) is the sentiment score on day $t$.

$$\Delta \text{ price}(t) = \text{price}(t) - \text{price}(t-1)$$

where price(t) is the price at time point $t$.

Next, determine the consistency of direction for each day.

$$\text{Consistency}(t) = \begin{cases} 1 & \text{if } (\Delta \text{ sentiment}(t) \times \Delta \text{ price}(t) > 0) \\ 0 & \text{otherwise} \end{cases}$$

$$\text{Consistency Percentage} = \frac{\sum_{t=1}^{n} \text{Consistency}(t)}{n} \times 100\%$$

To quantify the relationship between sentiment and stock price using the change rates and Pearson correlation coefficient we first need to calculate the change rate for both sentiment and price at each time point:

$$\text{ChangeRate}_{\text{sentiment}}(t) = \frac{\text{sentiment}(t) - \text{sentiment}(t-1)}{\text{sentiment}(t-1)}$$

$$\text{ChangeRate}_{\text{price}}(t) = \frac{\text{price}(t) - \text{price}(t-1)}{\text{price}(t-1)}$$

Next, we compute the Pearson correlation coefficient $r$ between these change rates:

$$r = \frac{\sum_{t=1}^{n} (\text{ChangeRate}_{\text{sentiment}}(t) - \overline{\text{ChangeRate}_{\text{sentiment}}})(\text{ChangeRate}_{\text{price}}(t) - \overline{\text{ChangeRate}_{\text{price}}})}{\sqrt{\sum_{t=1}^{n} (\text{ChangeRate}_{\text{sentiment}}(t) - \overline{\text{ChangeRate}_{\text{sentiment}}})^2} \sqrt{\sum_{t=1}^{n} (\text{ChangeRate}_{\text{price}}(t) - \overline{\text{ChangeRate}_{\text{price}}})^2}}$$

where $n$ is the total number of observations. To assess the accuracy of the stock price prediction model, we use the mean square error. The mean squared error (MSE) is a common evaluation metric for regression models, measuring the average squared difference between the estimated values and the actual value. The MSE is defined as:

$$\text{MSE} = \frac{1}{T} \sum_{i=1}^{T} (P_i - \hat{P}_i)^2$$

where $T$ denotes the total number of days in the forecast. $P_i$ represents the actual stock price on day i. $\widehat{P}_i$ represents the stock price predicted by the model on day i.

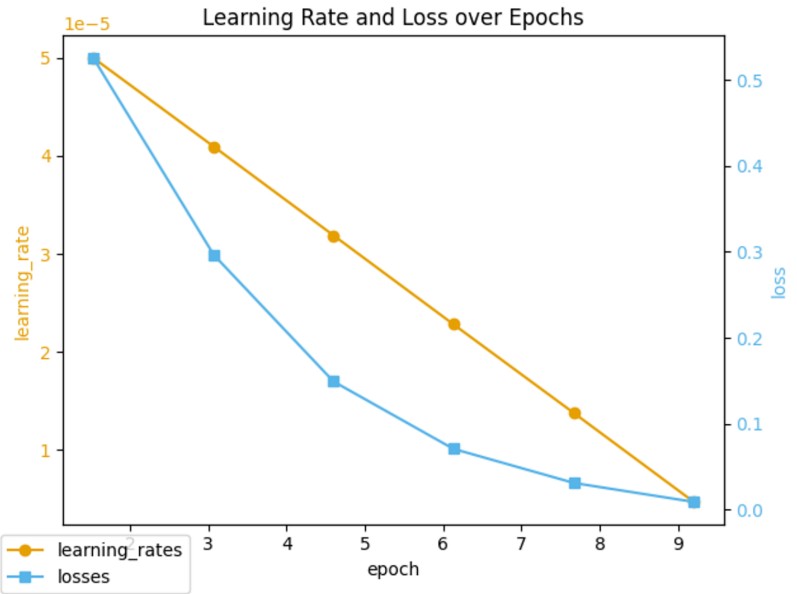

**Figure 5 Learning rate and loss during fine-tuning training.**

## Fine-tuning

Throughout the 10 epochs of fine-tuning, we closely monitored the learning rate and loss metrics, which are illustrated in Fig. 5. The initial learning rate was set to 5e−5 and gradually decreased as the epochs progressed. At the same time, the loss metric consistently declined, indicating an improvement in the model's ability to classify sentiment over time. To evaluate the efficacy of the fine-tuned model, we monitored performance metrics such as precision, recall, and F1 score, as shown in Fig. 6. These metrics were recorded regularly, providing valuable insights into the model's stability and adaptability. We observed that the fine-tuned model consistently outperformed the non-fine-tuned baseline, demonstrating significant improvements. In particular, the fine-tuned model showed notable enhancements in recall and F1 score, suggesting a more balanced approach to precision and generalizability in sentiment classification. A comparative analysis between the fine-tuned and non-fine-tuned models revealed substantial post-tuning improvements. These enhancements confirm our hypothesis that appropriately adjusted pre-trained models can effectively capture domain-specific sentiment nuances essential for financial analysis applications.

## Correlation between sentiment and stock

The word cloud shown in Fig. 7 is a visual representation of the frequency and importance of certain terms used in Twitter discussions related to TSLA. The keywords that appear with greater prominence, such as 'stock', 'market', 'buy', 'earnings', 'Elon Musk', and 'TSLA' suggest that these topics are highly discussed among Twitter users in the context of Tesla. The prominence of these terms indicates the public interest and sentiment surrounding Tesla as a company and as an investment option.

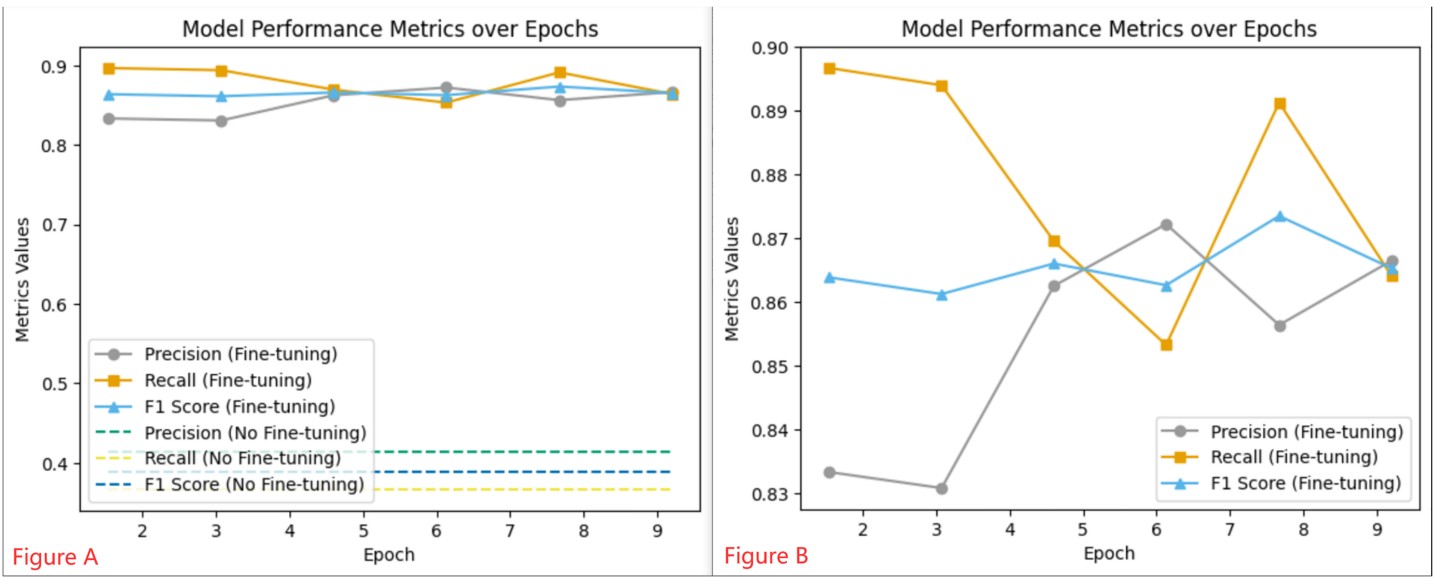

**Figure 6** Model Performance Metrics over Epochs: Graph (A) illustrates the consistent metrics of a Fine-tuning model *vs* the baseline of a non-Fine-tuning model, while Graph (B) details the dynamic changes in performance at various training checkpoints for the Fine-tuning model.

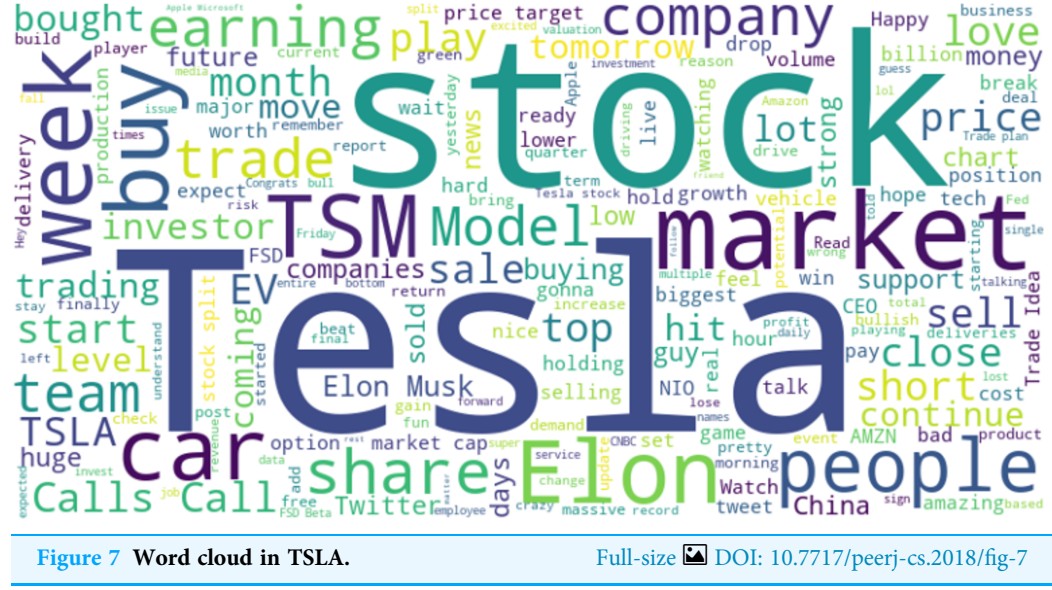

**Figure 7** Word cloud in TSLA.

Figures 8 and 9 display the average positive sentiment trend of TSLA and APPLE stocks respectively, overlaid on their stock prices over a specific period. A notable observation is that whenever there is an increase in positive sentiment on Twitter, there appears to be a corresponding uptick in the stock price. This trend is particularly evident in scenarios where positive sentiment spikes, suggesting that investor optimism, as reflected in Twitter discourse, may have a concurrent or predictive relationship with stock performance.

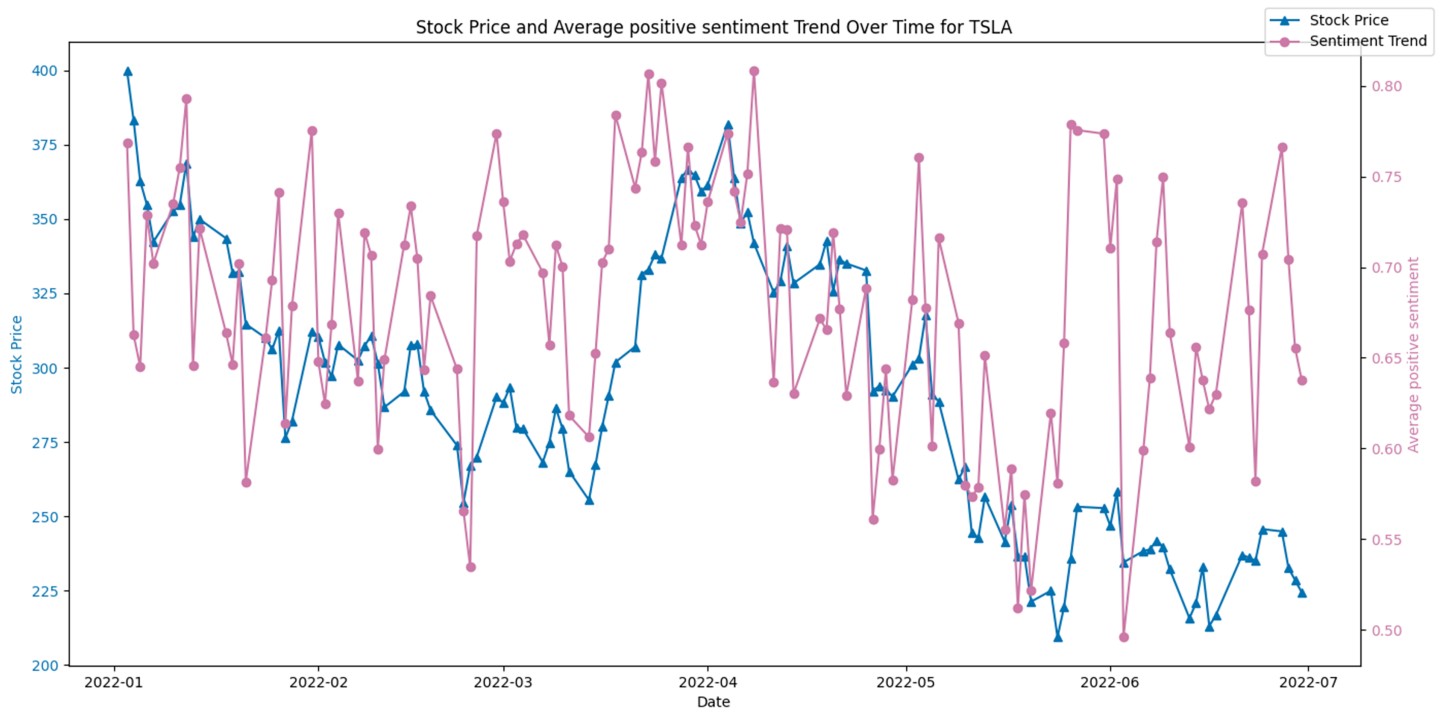

**Figure 8 Stock price and average positive sentiment trend over time for TSLA.**

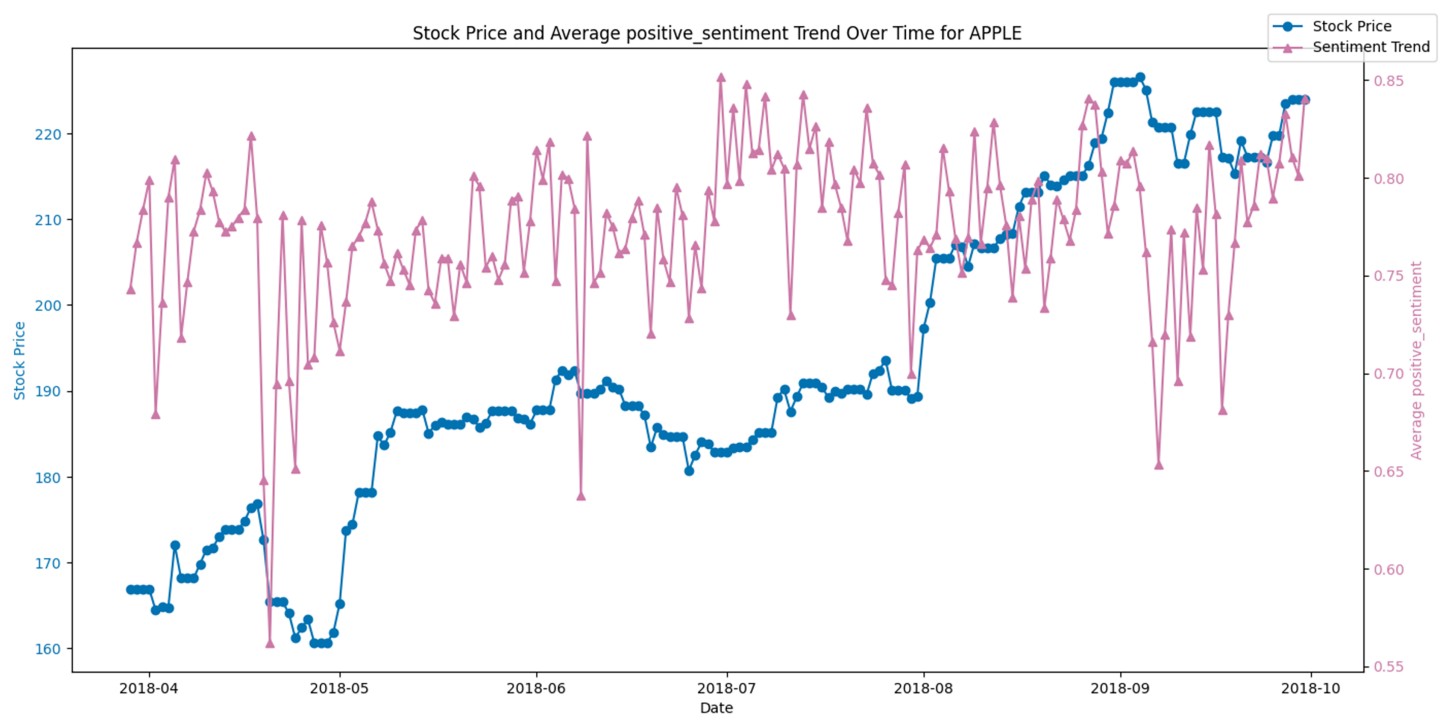

**Figure 9 Stock price and average positive sentiment trend over time for AAPL.**

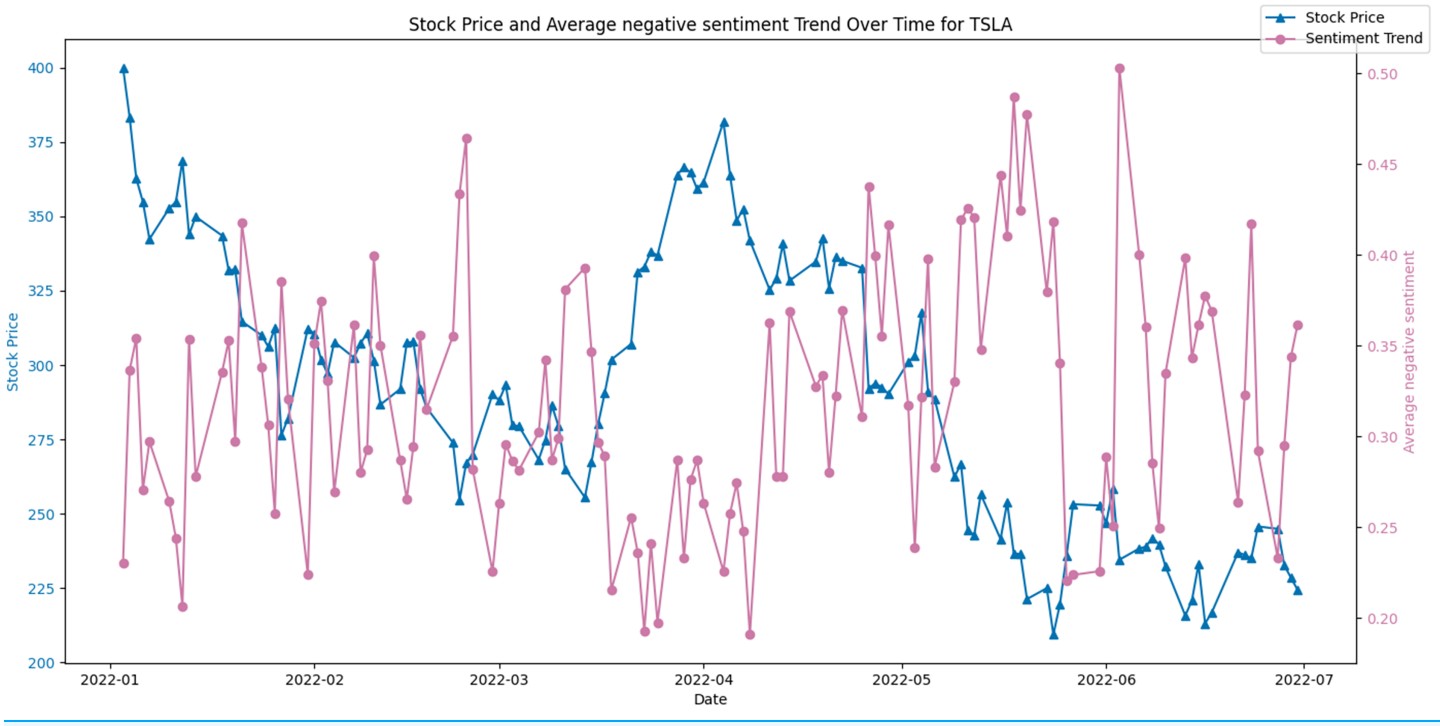

**Figure 10 Stock price and average negative sentiment Trend over time for TSLA.**

Conversely, Fig. 10 juxtaposes the stock price against the average negative sentiment trend. Here, an inverse relationship appears to be evident, where peaks in negative sentiment align with dips in the stock price. This potentially indicates that negative public perception could contribute to or signal a forthcoming decrease in the stock's value.

We have expanded our analysis by examining Pearson's correlation coefficient (r) and Consistency Percentage as metrics to evaluate the relationship between positive sentiment on Twitter and the stock market performance across various stocks. The results have been summarized in Tables 2 and 3. These tables present a comprehensive overview of tweet volumes, consistency of positive sentiment, and their respective correlation coefficients with stock performance for multiple companies over different periods. The 'tweet' column represents the volume of tweets related to each stock, which indicates the level of social media engagement and discussion intensity about the company. The 'Consistency Percentage' reflects the proportion of consistent positive sentiment over time. Higher percentages suggest a more uniformly positive public opinion toward the company, while lower percentages indicate greater sentiment fluctuation. This metric could be particularly useful for investors seeking to understand the stability of public perception around a stock. The Pearson's correlation coefficient (r) values range from positive to negative, suggesting varied strengths and directions of the linear relationship between stock performance and sentiment. A positive r value indicates that as positive sentiment increases, stock performance tends to increase, while a negative r value suggests that higher positive

**Table 2 Summary of Tweet volumes and correlation between public sentiment on social media and market performance.**

| Stock name | Date | Tweet | Consistency percentage | r |
|---|---|---|---|---|
| Tesla | From 2021-09-30 to 2022-01-01 | 21,562 | 62.50% | 0.33 |
| Tesla | From 2022-01-01 to 2022-06-30 | 43,361 | 72.36% | 0.55 |
| Tesla | From 2022-06-30 to 2022-09-30 | 16,230 | 52.38% | 0.21 |
| Tesla | From 2021-09-30 to 2022-09-30 | 80,793 | 65.74% | 0.43 |
| MSFT | From 2021-09-30 to 2022-01-01 | 785 | 43.75% | −0.05 |
| MSFT | From 2022-01-01 to 2022-06-30 | 2,468 | 51.22% | 0.13 |
| MSFT | From 2022-06-30 to 2022-09-30 | 857 | 57.14% | 0.34 |
| MSFT | From 2021-09-30 to 2022-09-30 | 4,089 | 51.39% | 0.06 |
| PG | From 2021-09-30 to 2022-01-01 | 785 | 48.44% | −0.04 |
| PG | From 2022-01-01 to 2022-06-30 | 2,468 | 46.34% | 0.08 |
| PG | From 2022-06-30 to 2022-09-30 | 857 | 47.62% | 0.00 |
| PG | From 2021-09-30 to 2022-09-30 | 4,089 | 48.21% | 0.01 |
| META | From 2021-09-30 to 2022-01-01 | 785 | 62.50% | 0.17 |
| META | From 2022-01-01 to 2022-06-30 | 1,915 | 56.03% | 0.04 |
| META | From 2022-06-30 to 2022-09-30 | 56 | 58.33% | 0.11 |
| META | From 2021-09-30 to 2022-09-30 | 2,751 | 58.72% | 0.05 |
| AMZN | From 2021-09-30 to 2022-01-01 | 785 | 51.56% | 0.02 |
| AMZN | From 2022-01-01 to 2022-06-30 | 2,468 | 56.10% | 0.20 |
| AMZN | From 2022-06-30 to 2022-09-30 | 857 | 58.73% | 0.27 |
| AMZN | From 2021-09-30 to 2022-09-30 | 4,089 | 55.78% | 0.08 |
| GOOG | From 2021-09-30 to 2022-01-01 | 225 | 55.36% | −0.13 |
| GOOG | From 2022-01-01 to 2022-06-30 | 779 | 42.62% | −0.03 |
| GOOG | From 2022-06-30 to 2022-09-30 | 290 | 55.17% | 0.08 |
| GOOG | From 2021-09-30 to 2022-09-30 | 1,291 | 49.79% | −0.02 |
| AMD | From 2021-09-30 to 2022-01-01 | 607 | 46.03% | −0.08 |
| AMD | From 2022-01-01 to 2022-06-30 | 1,177 | 57.85% | 0.25 |
| AMD | From 2022-06-30 to 2022-09-30 | 451 | 52.38% | 0.14 |
| AMD | From 2021-09-30 to 2022-09-30 | 2,227 | 54.03% | 0.18 |
| AAPL | From 2021-09-30 to 2022-01-01 | 1,191 | 51.56% | −0.13 |
| AAPL | From 2022-01-01 to 2022-06-30 | 2,426 | 55.28% | 0.11 |
| AAPL | From 2022-06-30 to 2022-09-30 | 1,460 | 66.67% | 0.34 |
| AAPL | From 2021-09-30 to 2022-09-30 | 5,056 | 57.37% | 0.13 |
| NFLX | From 2021-09-30 to 2022-01-01 | 232 | 61.02% | 0.14 |
| NFLX | From 2022-01-01 to 2022-06-30 | 1,207 | 47.32% | 0.03 |
| NFLX | From 2022-06-30 to 2022-09-30 | 291 | 50.91% | −0.15 |
| NFLX | From 2021-09-30 to 2022-09-30 | 1,727 | 51.98% | 0.04 |
| TSM | From 2021-09-30 to 2022-01-01 | 2,860 | 59.38% | 0.26 |
| TSM | From 2022-01-01 to 2022-06-30 | 6,115 | 53.66% | 0.11 |
| TSM | From 2022-06-30 to 2022-09-30 | 2,113 | 42.86% | −0.08 |
| TSM | From 2021-09-30 to 2022-09-30 | 11,034 | 52.59% | 0.07 |
| KO | From 2021-09-30 to 2022-01-01 | 43 | 43.48% | −0.30 |

| Stock name | Date | Tweet | Consistency percentage | r |
|---|---|---|---|---|
| KO | From 2022-01-01 to 2022-06-30 | 168 | 46.48% | 0.02 |
| KO | From 2022-06-30 to 2022-09-30 | 99 | 44.44% | 0.12 |
| KO | From 2021-09-30 to 2022-09-30 | 310 | 45.45% | −0.03 |
| F | From 2021-09-30 to 2022-01-01 | 4 | 33.33% | −0.37 |
| F | From 2022-01-01 to 2022-06-30 | 22 | 42.86% | −0.11 |
| F | From 2022-06-30 to 2022-09-30 | 6 | 40.00% | 0.11 |
| F | From 2021-09-30 to 2022-09-30 | 31 | 43.48% | −0.13 |
| COST | From 2021-09-30 to 2022-01-01 | 63 | 66.67% | 0.18 |
| COST | From 2022-01-01 to 2022-06-30 | 207 | 50.63% | 0.14 |
| COST | From 2022-06-30 to 2022-09-30 | 124 | 63.04% | −0.03 |
| COST | From 2021-09-30 to 2022-09-30 | 393 | 57.69% | 0.10 |

**Table 3 Summary of Tweet volumes and correlation between public sentiment on social media and market performance.**

| Stock name | Date | Tweet | Consistency percentage | r |
|---|---|---|---|---|
| DIS | From 2021-09-30 to 2022-01-01 | 193 | 61.54% | −0.02 |
| DIS | From 2022-01-01 to 2022-06-30 | 347 | 45.16% | −0.01 |
| DIS | From 2022-06-30 to 2022-09-30 | 102 | 54.05% | 0.19 |
| DIS | From 2021-09-30 to 2022-09-30 | 635 | 53.01% | 0.05 |
| VZ | From 2021-09-30 to 2022-01-01 | 33 | 57.89% | −0.34 |
| VZ | From 2022-01-01 to 2022-06-30 | 50 | 31.58% | −0.19 |
| VZ | From 2022-06-30 to 2022-09-30 | 41 | 52.00% | 0.04 |
| VZ | From 2021-09-30 to 2022-09-30 | 123 | 50.00% | −0.10 |
| CRM | From 2021-09-30 to 2022-01-01 | 58 | 64.29% | 0.07 |
| CRM | From 2022-01-01 to 2022-06-30 | 113 | 50.00% | −0.15 |
| CRM | From 2022-06-30 to 2022-09-30 | 62 | 24.14% | −0.05 |
| CRM | From 2021-09-30 to 2022-09-30 | 233 | 49.57% | −0.05 |
| INTC | From 2021-09-30 to 2022-01-01 | 60 | 37.50% | 0.15 |
| INTC | From 2022-01-01 to 2022-06-30 | 142 | 47.06% | −0.04 |
| INTC | From 2022-06-30 to 2022-09-30 | 115 | 51.28% | 0.05 |
| INTC | From 2021-09-30 to 2022-09-30 | 315 | 46.97% | 0.03 |
| BA | From 2021-09-30 to 2022-01-01 | 149 | 57.45% | 0.02 |
| BA | From 2022-01-01 to 2022-06-30 | 198 | 64.00% | 0.01 |
| BA | From 2022-06-30 to 2022-09-30 | 53 | 57.14% | 0.02 |
| BA | From 2021-09-30 to 2022-09-30 | 399 | 59.87% | 0.01 |
| BX | From 2021-09-30 to 2022-01-01 | 10 | 50.00% | 0.19 |
| BX | From 2022-01-01 to 2022-06-30 | 25 | 46.67% | 0.75 |
| BX | From 2022-06-30 to 2022-09-30 | 15 | 60.00% | −0.94 |
| BX | From 2021-09-30 to 2022-09-30 | 50 | 53.57% | −0.10 |
| NOC | From 2021-09-30 to 2022-01-01 | 4 | 33.33% | 0.40 |

(Continued)
| Table 3 (continued) | | | | |
|---|---|---|---|---|
| Stock name | Date | Tweet | Consistency percentage | r |
| NOC | From 2022-01-01 to 2022-06-30 | 22 | 42.86% | −0.25 |
| NOC | From 2022-06-30 to 2022-09-30 | 6 | 60.00% | 0.93 |
| NOC | From 2021-09-30 to 2022-09-30 | 31 | 47.83% | −0.15 |
| PYPL | From 2021-09-30 to 2022-01-01 | 287 | 47.37% | 0.05 |
| PYPL | From 2022-01-01 to 2022-06-30 | 463 | 57.61% | 0.14 |
| PYPL | From 2022-06-30 to 2022-09-30 | 95 | 51.43% | 0.08 |
| PYPL | From 2021-09-30 to 2022-09-30 | 843 | 53.51% | 0.10 |
| ENPH | From 2021-09-30 to 2022-01-01 | 53 | 21.05% | −0.06 |
| ENPH | From 2022-01-01 to 2022-06-30 | 40 | 55.00% | 0.30 |
| ENPH | From 2022-06-30 to 2022-09-30 | 124 | 60.00% | −0.08 |
| ENPH | From 2021-09-30 to 2022-09-30 | 216 | 51.16% | 0.04 |
| NIO | From 2021-09-30 to 2022-01-01 | 1,099 | 64.06% | 0.18 |
| NIO | From 2022-01-01 to 2022-06-30 | 1,491 | 54.47% | 0.23 |
| NIO | From 2022-06-30 to 2022-09-30 | 462 | 55.56% | 0.08 |
| NIO | From 2021-09-30 to 2022-09-30 | 3,021 | 58.17% | 0.20 |
| ZS | From 2021-09-30 to 2022-01-01 | 61 | 40.00% | 0.04 |
| ZS | From 2022-01-01 to 2022-06-30 | 99 | 44.44% | 0.01 |
| ZS | From 2022-06-30 to 2022-09-30 | 35 | 52.38% | −0.20 |
| ZS | From 2021-09-30 to 2022-09-30 | 193 | 47.52% | −0.03 |
| XPEV | From 2021-09-30 to 2022-01-01 | 116 | 66.67% | 0.10 |
| XPEV | From 2022-01-01 to 2022-06-30 | 100 | 48.65% | −0.24 |
| XPEV | From 2022-06-30 to 2022-09-30 | 16 | 54.55% | 0.23 |
| XPEV | From 2021-09-30 to 2022-09-30 | 225 | 60.87% | −0.14 |

sentiment is associated with a decrease in stock performance. We can find that the r-value is also relatively high when the number of Twitter comments is sufficiently high. For example, 'TSLA' has the highest number of comments, and the sentiment of its comments is highly correlated with the stock price.

## Price forecasts

In Fig. 11, the predicted data closely follows the actual data, suggesting that the model has a reasonable level of predictive power. The model appears to capture the general trend of the stock prices, although some deviations are present, indicating areas where the model could be refined. Notably, the model seems to have some limitations in capturing sharp spikes or drops in the stock price, which could be due to abrupt market movements that are not immediately reflected in sentiment scores.

In the case of Tesla as shown in Fig. 12, the predicted data appears to deviate less from the actual data when compared to AAPL, possibly due to Tesla's stock being subject to more abrupt price changes influenced by factors beyond public sentiment, such as executive decisions, technological advancements, and regulatory news.

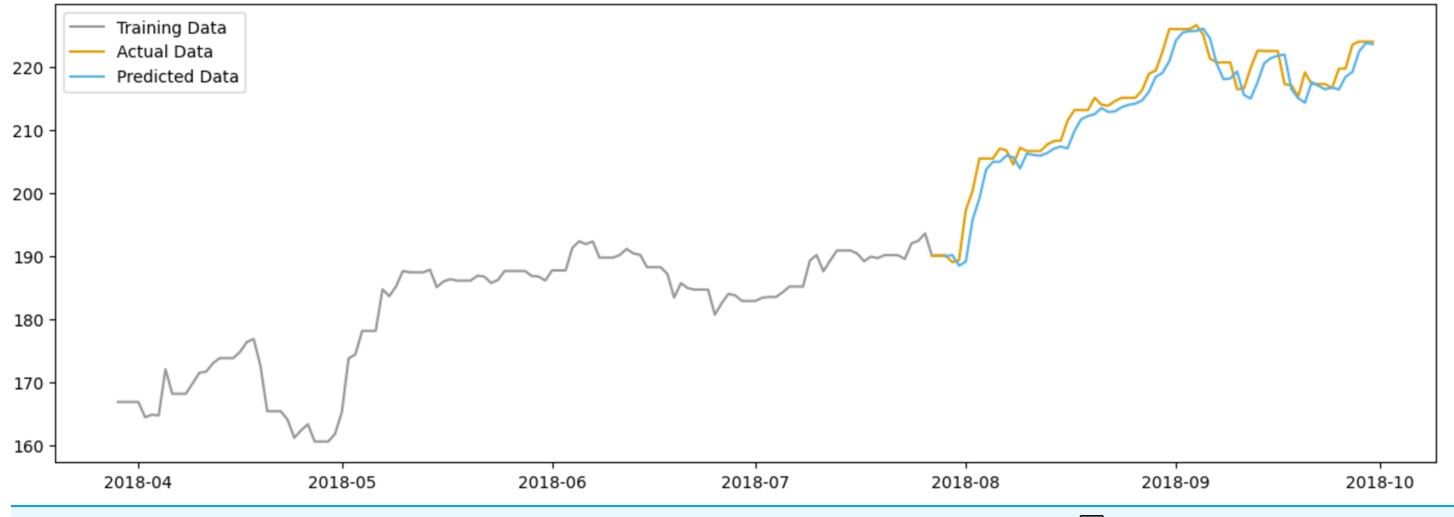

**Figure 11 Comparison between the predicted and actual stock prices of AAPL.**

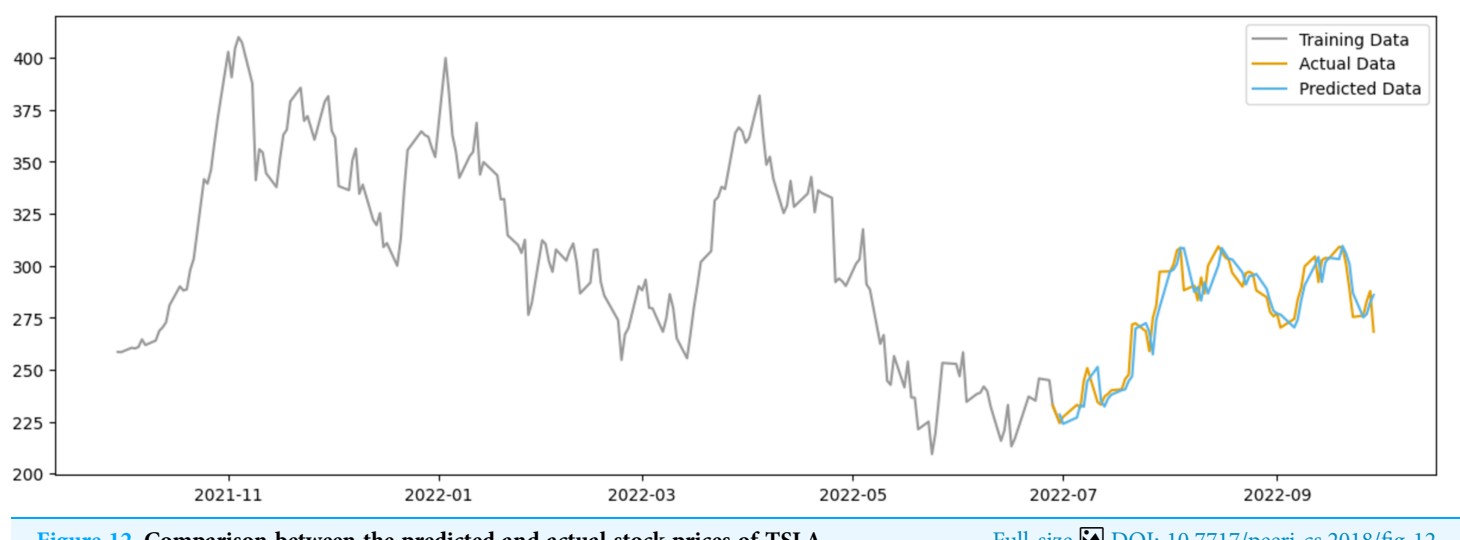

**Figure 12 Comparison between the predicted and actual stock prices of TSLA.**

However, both figures suggest that while positive sentiment is a valuable predictor, it should be one of multiple factors considered in a comprehensive stock price prediction model. Additional variables that could be integrated into future models include market trends, economic indicators, and company-specific news.

## FUTURE WORK

Given the limited exploration of the long-term impact of social media sentiment on stock markets, future work could focus on extending the analysis to assess the sustainability and long-term reliability of using social media data for financial forecasting. This could involve studying the influence of sentiment over extended periods and during unprecedented

events or market shifts. Considering the potential limitations of relying solely on social media sentiment for stock price predictions, future work could explore the integration of additional data sources such as market trends, economic indicators, and company-specific news. This multi-faceted approach could lead to more comprehensive and accurate stock price prediction models.

### Funding
The authors received no funding for this work.

### Competing Interests
The authors declare that they have no competing interests.

### Author Contributions
- Kaifeng Guo conceived and designed the experiments, performed the experiments, analyzed the data, performed the computation work, prepared figures and/or tables, authored or reviewed drafts of the article, and approved the final draft.
- Haoling Xie analyzed the data, prepared figures and/or tables, authored or reviewed drafts of the article, and approved the final draft.

### Data Availability
The code is available at GitHub and Zenodo.

- https://github.com/w44607797/Deep-Learning-in-Finance-Assessing-Twitter-Sentiment-Impact-and-Prediction-on-Stocks.

- Guo, K., & Xie, H. (2024). Deep learning in finance assessing twitter sentiment impact and prediction on stocks. Zenodo. https://doi.org/10.5281/zenodo.10825422.

The Stock-Market Sentiment Dataset is available at Kaggle: https://www.kaggle.com/datasets/yash612/stockmarket-sentiment-dataset/data, Yash Chaudhary, DOI 10.34740/kaggle/dsv/1217821.

The Stock Tweets for Sentiment Analysis and Prediction dataset is available at Kaggle: https://www.kaggle.com/datasets/equinxx/stock-tweets-for-sentiment-analysis-and-prediction.

The Tweets about the Top Companies from 2015 to 2020 dataset is available at Kaggle:

- https://www.kaggle.com/datasets/omermetinn/tweets-about-the-top-companies-from-2015-to-2020/data?select=Company.csv.

- https://www.kaggle.com/datasets/omermetinn/values-of-top-nasdaq-copanies-from-2010-to-2020.

### Supplemental Information
Supplemental information for this article can be found online at http://dx.doi.org/10.7717/peerj-cs.2018#supplemental-information.

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
