# Peer review of "Deep learning in finance assessing twitter sentiment impact and prediction on stocks"

_PeerJ Computer Science, doi:10.7717/peerj-cs.2018_

## Round 0.1 · original submission · Major Revisions

Your paper has been assessed by our reviewers. Please see the attached reviewer comments for further details about necessary revisions.

A point-by-point response letter must accompany your revised manuscript. This letter must provide a detailed response to each reviewer/editorial point raised, describing exactly what amendments have been made to the manuscript text and where these can be viewed (e.g. Methods section, line 12, page 5). If you disagree with any comments raised, please provide a detailed rebuttal to help explain and justify your decision.

**Language Note:** The review process has identified that the English language must be improved. PeerJ can provide language editing services - please contact us at [email protected] for pricing (be sure to provide your manuscript number and title). Alternatively, you should make your own arrangements to improve the language quality and provide details in your response letter. – PeerJ Staff

Reviewer 1 ·

Basic reporting

The authors' strategy was to focus on optimizing the developed sentiment extraction models to improve their accuracy and robustness. The article is well-written and provides a comprehensive overview of the approach. However, I would suggest adding a dedicated "Future Work" section in the article. This section could be valuable for discussing possible research directions and areas where the work could be extended or improved. This would allow readers to gain a clearer understanding of the future prospects of the research. It is recommended to organize the figures in the order they are cited in the paper, especially in the 'Experimental Setting' section, specifying the figure to which reference is made.
Furthermore, to further enhance the article, the authors might consider adding case studies or practical examples in other domains.

Experimental design

The authors, in this paper, aimed to quantify the correlation between transient sentiments expressed on social media and measurable fluctuations in the stock market. They adopted an existing sentiment analysis algorithm, refining it to create a specific model for evaluating sentiment in tweets associated with financial markets. The model was trained and validated using a comprehensive dataset of stock-related discussions on Twitter, enabling the identification of subtle emotional cues that could predict changes in stock prices. The approach employed is quantitative, and methodical tests revealed a statistically significant relationship between sentiment expressed on Twitter and subsequent stock market activity. These findings suggest that machine learning algorithms can be instrumental in enhancing the analytical capabilities of financial experts.

Validity of the findings

The optimized model consistently demonstrated superior performance compared to the non-optimized reference model, highlighting significant improvements. Specifically, the optimized model showed a notable enhancement in recall and F1 score, suggesting a more balanced approach to precision and generalizability in sentiment classification. Comparative analysis between the optimized and non-optimized models revealed significant improvements post-optimization. This finding confirms the hypothesis that appropriately adapted pre-trained models can effectively capture nuances of domain-specific sentiment, crucial for applications in financial analysis.

Additional comments

No comment

Reviewer 2 ·

Basic reporting

The contribution consists of refining a model that evaluates sentiment on tweets related to the stock market to predict changes in stock prices using a dataset of stock-related discussions on Twitter and revealing significant relationship between sentiment and stock market.

Figures and Tables are adequate. The legend of Figure 4 should include a brief explanation about what the red and blue curves stand for. Figure 10 should include a legend to specify what the horizontal axis refers to. Figure 11 should include legends to specify what the horizontal and vertical axes refer to. Is it return or price? Because of the range of values on the vertical axis, I guess it refers to stock price. Some legends for axes should be included in Figure 4. The same comments apply to Figure 12. Character size in Figures is small and hard to read.

In general, the manuscript is well written; however, there are some typos and some minor errors to adjust. The last paragraph in page 8 does not start with a capital character. Changes to be made: "These tabels" -> "These tables".

It is always helpful to find the mathematical description of the regression model used for this application. Figure 1 explains the machine learning model is an RNN; however, there is some information missing, specifically the parameters of the learning algorithm.

The references are adequate and recent.

Experimental design

In general, the experimental set up is well explained. There are some omissions regarding the training algorithm.

Most of the employed methods are well known and the contribution is the integration of methods. A brief description of a method as well as references are handful for some readers.

Validity of the findings

The results depicted in Figure 11 and 12 are very promising. Are they the best qualitative results?

It is of interest to discover if sentiment form twiter is correlated with the stock prices, which makes this research valuable.

Additional comments

The manuscript deserves publication after major revision.

---

## Round 0.2 · accepted · Accept

Based on the reviewers' reports, and my own assessment as Editor, I am pleased to inform you that the manuscript is acceptable for publication in PeerJ Computer Science.

Reviewer 1 ·

Basic reporting

The authors' strategy was to focus on optimizing the developed sentiment extraction models to improve their accuracy and robustness. The article is well-written and provides a comprehensive overview of the approach. However, I would suggest adding a dedicated "Future Work" section in the article. This section could be valuable for discussing possible research directions and areas where the work could be extended or improved. This would allow readers to gain a clearer understanding of the future prospects of the research. It is recommended to organize the figures in the order they are cited in the paper, especially in the 'Experimental Setting' section, specifying the figure to which reference is made.
Furthermore, to further enhance the article, the authors might consider adding case studies or practical examples in other domains.

Experimental design

The authors, in this paper, aimed to quantify the correlation between transient sentiments expressed on social media and measurable fluctuations in the stock market. They adopted an existing sentiment analysis algorithm, refining it to create a specific model for evaluating sentiment in tweets associated with financial markets. The model was trained and validated using a comprehensive dataset of stock-related discussions on Twitter, enabling the identification of subtle emotional cues that could predict changes in stock prices. The approach employed is quantitative, and methodical tests revealed a statistically significant relationship between sentiment expressed on Twitter and subsequent stock market activity. These findings suggest that machine learning algorithms can be instrumental in enhancing the analytical capabilities of financial experts.

Validity of the findings

The optimized model consistently demonstrated superior performance compared to the non-optimized reference model, highlighting significant improvements. Specifically, the optimized model showed a notable enhancement in recall and F1 score, suggesting a more balanced approach to precision and generalizability in sentiment classification. Comparative analysis between the optimized and non-optimized models revealed significant improvements post-optimization. This finding confirms the hypothesis that appropriately adapted pre-trained models can effectively capture nuances of domain-specific sentiment, crucial for applications in financial analysis.

Additional comments

The given instructions have been mostly met; however, inconsistencies have been noted in the positioning of the figures. Specifically, some of them appear to be placed adjacent to the references, compromising the clarity and readability of the text. I would suggest carefully reassessing the arrangement of these figures and ensuring they are properly integrated into the text while adhering to the formatting guidelines prescribed by the publication.

Reviewer 2 ·

Basic reporting

The authors have updated the original manuscript according to the reviewer comments. The manuscript is now ready to be published.

Experimental design

Most of the reviewer observations were related to writing and findings. There was one comment regarding quality vs. quantity that the authors have considered.

Validity of the findings

The authors have updated the original manuscript according to the reviewer comments. The manuscript is now ready to be published.

Additional comments

The paper is ready to be published.